# VIDEO ACTION SEGMENTATION WITH HYBRID TEMPORAL NETWORKS

## ABSTRACT

Action segmentation as a milestone towards building automatic systems to understand untrimmed videos has received considerable attention in the recent years. It is typically being modeled as a sequence labeling problem but contains intrinsic and sufficient differences than text parsing or speech processing. In this paper, we introduce a novel hybrid temporal convolutional and recurrent network (TricorNet), which has an encoder-decoder architecture: the encoder consists of a hierarchy of temporal convolutional kernels that capture the local motion changes of different actions; the decoder is a hierarchy of recurrent neural networks that are able to learn and memorize long-term action dependencies after the encoding stage. Our model is simple but extremely effective in terms of video sequence labeling. The experimental results on three public action segmentation datasets have shown that the proposed model achieves superior performance over the state of the art.

## 1 INTRODUCTION

Action segmentation is a challenging problem in high-level video understanding. In its simplest form, action segmentation aims to segment a temporally untrimmed video by time and label each segmented part with one of $k$ pre-defined action labels. For example, given a video of *Making Hotdog* (see Fig. 1), we label the first 10 seconds as *take bread*, and the next 20 seconds as *take sausage*, and the remaining video as *pour ketchup* following the procedure dependencies of making a hotdog. The results of action segmentation can be further used as input to various applications, such as video-to-text (Das et al., 2013) and action localization (Mettes et al., 2016).

Most current approaches for action segmentation (Yeung et al., 2015; Singh et al., 2016a; Huang et al., 2016) use features extracted by convolutional neural networks, e.g., two-stream CNNs (Simonyan & Zisserman, 2014) or local 3D ConvNets (Tran et al., 2015), at every frame after a downsampling as the input, and apply a one-dimensional sequence prediction model, such as recurrent neural networks, to label actions on frames. Despite the simplicity in handling video data, action segmentation is treated similar to text parsing (Cross & Huang, 2016), which results the local motion changes in various actions being under-explored. For example, the action *pour ketchup* may consist of a series of sub-actions, e.g., *pick up the ketchup*, *squeeze and pour*, and *put down the ketchup*. Furthermore, the time duration of performing the same action *pour ketchup* may vary according to different people and contexts.

Indeed, the recent work by Lea et al. (2017) starts to explore the local motion changes in action segmentation. They propose an encoder-decoder framework, similar to the deconvolution networks in image semantic segmentation (Noh et al., 2015), for video sequence labeling. By using a hierarchy of 1D temporal convolutional and deconvolutional kernels in the encoder and decoder networks, respectively, their model is effective in terms of capturing the local motions and achieves state-of-the-art performance in various action segmentation datasets. However, one obvious drawback is that it fails to capture the long-term dependencies of different actions in a video due to its fixed-size, local receptive fields. For example, *pour ketchup* usually happens after both *take bread* and *take sausage* for a typically video of *Making Hotdog*. In addition, a dilated temporal convolutional network, similar to the WavNet for speech processing (van den Oord et al., 2016), is also tested in (Lea et al., 2017), but has worse performance, which further suggests the existence of differences between video and speech data, despite they are both being represented as sequential features.

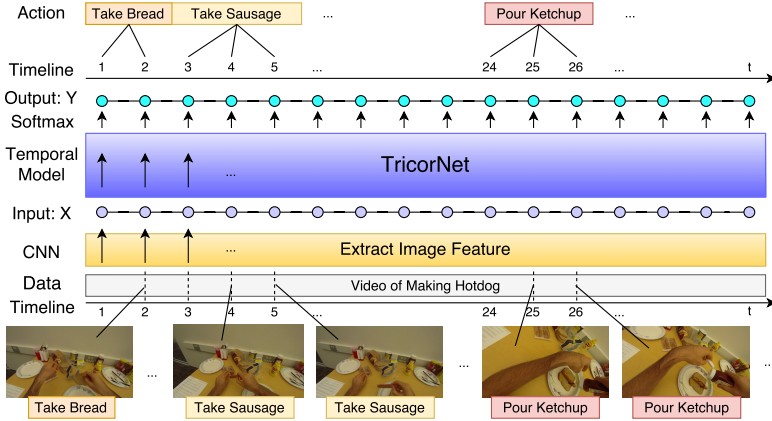

Figure 1: An overview of video action segmentation problems with our proposed methodology, with example video and action labels from GTEA Fathi et al. (2011) dataset.

To overcome the above limitations, we propose a novel hybrid TempoRal COnvolutional and Recurrent Network (TricorNet), that attends to both local motion changes and long-term action dependencies for modeling video action segmentation. TricorNet uses frame-level features as the input to an encoder-decoder architecture. The encoder is a temporal convolutional network that consists of a hierarchy of one-dimensional convolutional kernels, observing that the convolutional kernels are good at encoding the local motion changes; the decoder is a hierarchy of recurrent neural networks, in our case Bi-directional Long Short-Term Memory networks (Bi-LSTMs) (Graves et al., 2005), that are able to learn and memorize long-term action dependencies after the encoding process. Our network is simple but extremely effective in terms of dealing with different time durations of actions and modeling the dependencies among different actions.

We conduct extensive experiments on three public action segmentation datasets, where we compare our proposed models with a set of recent action segmentation networks using three different evaluation metrics. The quantitative experimental results show that our proposed TricorNet achieves superior or competitive performance to state of the art on all three datasets. A further qualitative exploration on action dependencies shows that our model is good at capturing long-term action dependencies and produce smoother labeling.

For the rest of the paper, we first survey related work in the domain of action segmentation and action detection in Sec. 2. We introduce our hybrid temporal convolutional and recurrent network with some implementation variants in Sec. 3. We present both quantitative and qualitative experimental results in Sec. 4, and conclude the paper in Sec. 5.

## 2 RELATED WORK

For action segmentation, many existing works use frame-level features as the input and then build temporal models on the whole video sequence. Yeung et al. (2015) propose an attention LSTM network to model the dependencies of the input frame features in a fixed-length window. Huang et al. (2016) consider the weakly-supervised action labeling problem. Singh et al. (2016a) present a multi-stream bi-directional recurrent neural network for fine-grained action detection task. Lea et al. (2017) propose two temporal convolutional networks for action segmentation and detection. The design of our model is inspired by Singh et al. (2016a) and Lea et al. (2017) and we compare with them in the experiments.

Lea et al. (2016a) introduce a spatial CNN and a spatiotemporal CNN; the latter is an end-to-end approach to model the whole sequence from frames. Here, we use the output features of their spatial CNN as the input to our TricorNet, and compare with their results obtained by the spatiotemporal CNN. Richard & Gall (2016) use a statistical language model to focus on localizing and classifying segments in videos of varying lengths. Kuehne et al. (2016) propose an end-to-end generative

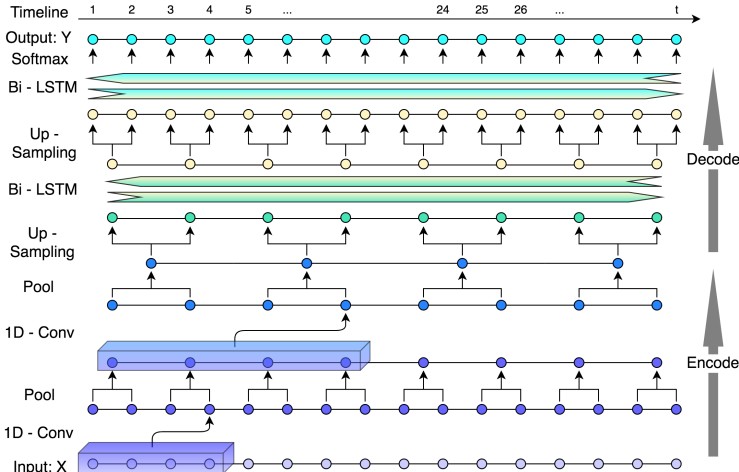

Figure 2: The overall framework of our proposed TricorNet. The encoder network consists of a hierarchy of temporal convolutional kernels that are effective at capturing local motion changes. The decoder network consists of a hierarchy of Bi-LSTMs that model the long-term action dependencies.

approach for action segmentation, using Hidden Markov Models on dense trajectory features. We also compare with their results on the 50 Salads dataset.

Another related area is action detection. Peng & Schmid (2016) propose a two-stream R-CNN to detect actions. Yeung et al. (2016) introduce an action detection framework based on reinforcement learning. Li et al. (2016) present joint classification-regression recurrent neural networks for human action detection from 3D skeleton data. Those methods primarily work for single-action, short videos. The recent work by Zhou et al. (2017) considers action detection and dependencies for untrimmed and unconstrained Youtube videos. It is likely that the action segmentation and detection works if fused can be beneficial to each other, but the topic is out of the scope of this paper.

## 3 MODEL

In this section, we introduce our proposed model along with some implementation details. The input to our TricorNet is a set of frame-level video features, e.g., output from a CNN, for each frame of a given video. Let $X_t$ be the input feature vector at time step $t$ for $1 \leq t \leq T$, where $T$ is the total number of time steps in a video sequence that may vary among videos in a dataset. The action label for each frame is defined by a sparse vector $Y_t \in \{0, 1\}^c$, where $c$ is the number of classes, such that the true class is $1$ and all others are $0$. In case there are frames that do not have a pre-defined action label, we use one additional *background* label for those frames.

### 3.1 TEMPORAL CONVOLUTIONAL AND RECURRENT NETWORK

A general framework of our TricorNet is depicted in Fig. 2. The TricorNet has an encoder-decoder structure. Both encoder and decoder networks consist of $K$ layers. We define the encoding layer as $L_E^{(i)}$, and the decoding layer as $L_D^{(i)}$, for $i = 1, 2, \ldots, K$. There is a middle layer $L_{mid}$ between the encoder and the decoder. Here, $K$ is a hyper-parameter that can be turned based on the size and appearance of the video data in a dataset. A large $K$ means the network is deep and, typically, requires more data to train. Empirically, we set $K = 2$ for all of our experiments.

In the encoder network, each layer $L_E^{(i)}$ is a combination of temporal (1D) convolutions, a non-linear activation function $E = f(\cdot)$, and max pooling across time. Using $F_i$ to specify the number of convolutional filters in an encoding layer $L_E^{(i)}$, we define the collection of convolutional filters as $W_E^{(i)} = \{W^{(j)}\}_{j=1}^{F_i}$ with a corresponding bias vector $b_E^i \in \mathbb{R}^{F_i}$. Given the output from the previous

encoding layer after pooling $E^{(i-1)}$, we compute activations of the current layer $L_E^{(i)}$:

$$E^{(i)} = f(W_E^{(i)} * E^{(i-1)} + b_E^{(i)}) \ , \tag{1}$$

where $*$ denotes the 1D convolution operator. Note that $E^{(0)} = (X_1, \ldots, X_T)$ is the collection of input frame-level feature vectors. The length of the convolutional kernel is another hyper-parameter. A larger length means a larger receptive field, but will also reduce the discrimination between adjacent time steps. We report the best practices in Sec. 4.

Here, the middle level layer $L_{mid}$ is the output of the last encoding layer $E^{(K)}$ after the pooling, and it is used as the input to the decoder network. The structure of the decoding part is a reserved hierarchy compared to the encoding part, and it also consists of $K$ layers. We use Bi-directional Long Short-Term Memory (Bi-LSTM) (Graves et al., 2005) units to model the long-range action dependencies, and up-sampling to decode the frame-level labels. Hence, each layer $L_D^{(i)}$ in the decoder network is a combination of up-sampling and Bi-LSTM.

Generally, recurrent neural networks use a hidden state representation $h = (h_1, h_2, \ldots, h_t)$ to map the input vector $x = (x_1, x_2, \ldots, x_t)$ to the output sequence $y = (y_1, y_2, \ldots, y_t)$. In terms of the LSTM unit, it updates its hidden state by the following equations:

$$
\begin{aligned}
i_t &= \sigma(W_{xi}x_t + W_{hi}h_{t-1} + b_i) \ , \\
f_t &= \sigma(W_{xf}x_t + W_{hf}h_{t-1} + b_f) \ , \\
o_t &= \sigma(W_{xo}x_t + W_{ho}h_{t-1} + b_o) \ , \\
g_t &= \tanh(W_{xc}x_t + W_{hc}h_{t-1} + b_c) \ , \\
c_t &= f_t c_{t-1} + i_t g_t \ , \\
h_t &= o_t \tanh(c_t) \ ,
\end{aligned}
\tag{2}
$$

where $\sigma(\cdot)$ is a sigmoid activation function, $\tanh(\cdot)$ is the hyperbolic tangent activation function, $i_t$, $f_t$, $o_t$, and $c_t$ are the input gate, forget gate, output gate, and memory cell activation vectors, respectively. Here, $W$s and $b$s are the weight matrices and bias terms. A Bi-LSTM layer contains two LSTMs: one goes forward through time and one goes backward. The output is a concatenation of the results from the two directions.

In TricorNet, we use the updated sequences of hidden states $H^{(i)}$ as the output of each decoding layer $L_D^{(i)}$. We use $H_i$ to specify the number of hidden states in a single LSTM layer. Hence, for layer $L_D^{(i)}$, the output dimension at each time step will be $2H_i$ as a concatenation of a forward and a backward LSTM. The output of the whole decoding part will be a matrix $D = H^{(K)} \in \mathbb{R}^{T \times 2H_K}$, which means at each time step $t = 1, 2, \ldots, T$, we have a $2H_K$-dimension vector $D_t$ that is the output of the last decoding layer $L_D^{(K)}$.

Finally, we have a softmax layer across time to compute the probability that the label of frame at each time step $t$ takes one of the $c$ action classes, which is given by:

$$\hat{Y}_t = softmax(W_d D_t + b_d) \ , \tag{3}$$

where $\hat{Y}_t \in [0, 1]^c$ is the output probability vector of $c$ classes at time step $t$, $D_t$ is the output from our decoder for time step $t$, $W_d$ is a weight matrix and $b_d$ is a bias term.

### 3.2 MODEL VARIATION

In order to find the best structure of combining temporal convolutional layers and Bi-LSTM layers, we test three different model designs (shown in Fig. 3). We detail their architectures as follows, and the experiments of different models are presented in Sec. 4.

**TricorNet.** TricorNet uses temporal convolutional kernels for encoding layers $L_E^{(i)}$ and uses Bi-LSTMs for decoding layers $L_D^{(i)}$, as proposed in Fig. 2. The intuition of this design is to use different temporal convolutional layers to encode local motion changes, and apply different Bi-LSTM layers to decode the sequence and learn different levels of long-term action dependencies.

**TricorNet (high).** TricorNet (high) deploys the Bi-LSTM units only at the middle-level layer $L_{mid}$, which is the layer between encoder and decoder. It uses temporal convolutional kernels for

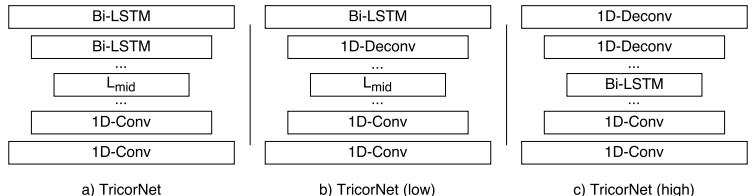

Figure 3: Model variants of TricorNet.

both encoding layers $L_E^{(i)}$ and decoding layers $L_D^{(i)}$. The intuition is to use Bi-LSTM to model the sequence dependencies at an abstract level, where the information is highly compressed while keeping both encoding and decoding locally. This is expected to perform well when action labels are coarse.

**TricorNet (low).**  TricorNet (low) deploys the Bi-LSTM units only at the layer $L_D^{(K)}$, which is the last layer of the decoder. It uses temporal convolutional kernels for the encoding layers $L_E^{(i)}$ and the rest of the layers $L_D^{(i)}$, where $i < k$ in the decoder. The intuition of this design is to use Bi-LSTM to decode details only. For the cases where action labels are fine-grained, it is better to focus on learning dependencies at a low-level, where information is less compressed.

### 3.3 Implementation Details

In this work, some of the hyper-parameters of all three TricorNets are fixed and used throughout all the experiments. In the encoding part, we use max pooling with width $= 2$ across time. Each temporal convolutional layer $L_E^{(i)}$ has $32 + 32i$ filters. In the decoding part, up-sampling is performed by simply repeating each entry twice. The latent state of each LSTM layer $L_D^{(i)}$ is given by $2H_i$. We use Normalized Rectified Linear Units (Lea et al., 2017) as the activation function for all the temporal convolutional layer, which is defined as:

$$Norm.ReLU(\cdot) = \frac{ReLU(\cdot)}{\max\big(ReLU(\cdot)\big) + \epsilon} \ , \tag{4}$$

where $\max\big(ReLU(\cdot)\big)$ is the maximal activation value in the layer and $\epsilon = 10^{-5}$.

In our experiments, the models are trained from scratch using only the training set of the target dataset. Weights and parameters are learned using the categorical cross entropy loss with Stochastic Gradient Descent and ADAM (Kingma & Ba, 2014) step updates. We also add spatial dropouts between convolutional layers and dropouts between Bi-LSTM layers. The models are implemented with Keras (Chollet, 2015) and TensorFlow.

## 4 Experimental Results

We conduct quantitative experiments on three challenging action segmentation datasets and use three different metrics to evaluate the performance of TricorNet variants. For each dataset, we compare our results with some baseline methods as well as state of the art results.

### 4.1 Datasets

University of Dundee 50 Salads (Stein & McKenna, 2013) dataset captures 25 people preparing mixed salad two times each, and has annotated accelerometer and RGB-D video data. In our experiment, we only use the features extracted from video data. The duration of videos varies from 5 to 10 minutes. We use the fine-granularity action labels (*mid-level*), where each video contains 17 classes of actions performed when making salads, such as *cut cheese*, *peel cucumber*. We use the spatial CNN (Lea et al., 2016a) features as our input with cross validation on five splits, which are provided by Lea et al. (2017).

Georgia Tech Egocentric Activities (GTEA) (Fathi et al., 2011) dataset contains seven types of daily activities such as making sandwich, tea, or coffee. Each activity is performed by four different

Table 1: Experimental results on University of Dundee 50 Salads dataset. We use frame-wise accuracy (Acc.), segmental edit score (Edit) and overlap F1 score with thresholds (F1@10, 25, 50) for evaluation. The top-two results for each metric are in boldface; the same applies for other tables.

| 50 Salads (Mid) | Acc. | Edit | F1@{10, 25, 50} |
|---|---|---|---|
| Spatial CNN (Lea et al., 2016a) | 54.9 | 24.8 | 32.3, 27.1, 18.9 |
| IDT+LM (Richard & Gall, 2016) | 48.7 | 45.8 | 44.4, 38.9, 27.8 |
| ST-CNN (Lea et al., 2016a) | 59.4 | 45.9 | 55.9, 49.6, 37.1 |
| Bi-LSTM | 55.7 | 55.6 | 62.6, 58.3, 47.0 |
| Dilated TCN (Lea et al., 2017) | 59.3 | 43.1 | 52.2, 47.6, 37.4 |
| ED-TCN (Lea et al., 2017) | 64.7 | **59.8** | **68.0**, **63.9**, 52.6 |
| TricorNet (high) | **65.5** | 59.3 | 66.4, 62.7, **53.3** |
| TricorNet (low) | 65.4 | 59.0 | 67.0, 63.2, 52.1 |
| TricorNet | **67.5** | **62.8** | **70.1**, **67.2**, **56.6** |

people, thus totally 28 videos. For each video, there are about 20 fine-grained action instances such as *take bread*, *pour ketchup*, in approximately one minute. To make the results comparable, we use the features provided by Lea et al. (2017), which are outputs from a spatial CNN, also with cross-validation splits.

JHU-ISI Gesture and Skill Assessment Working Set (JIGSAWS) (Gao et al., 2014) consists of 39 videos capturing eight surgeons performing elementary surgical tasks on a bench-top. Each surgeon has about five videos with around 20 action instances. There are totally 10 different action classes. Each video is around two minutes long. In this work, we use the videos of suturing tasks. Features and cross-validation splits are provided by Lea et al. (2016c).

## 4.2 METRICS

One of the most common metrics used in action segmentation problems is frame-wise accuracy, which is straight-forward and can be computed easily. However, a drawback is that models achieving similar accuracy may have large qualitative differences. Furthermore, it fails to handle the situation that models may produce large over-segmentation errors but still achieve high frame-wise accuracy. As a result, some work (Lea et al., 2016a;b) tends to use a segmental edit score, as a complementary metric, which penalizes over-segmentation errors. Recently, Lea et al. (2017) use a segmental overlap F1 score, which is similar to mean Average Precision (mAP) with an Intersection-Over-Union (IoU) overlap criterion. In this paper, we use frame-wise accuracy for all three datasets. According to reported scores by other papers, we also use segmental edit score for JIGSAWS, overlap F1 score with threshold $k = 10, 25, 50$ for GTEA, and for 50 Salads we use both.

## 4.3 RESULTS

In experiments, we have tried different convolution lengths and number of layers on each dataset; we report the best practices in discussions below. A *background* label is introduced for frames that do not have an action label. Since we are using the same features as Lea et al. (2017) for 50 Salads and GTEA, and features from Lea et al. (2016c) for JIGSAWS, we also obtain the results of baselines from their work along with state-of-the-art methods.

**50 Salads.** In Table 1 , we show that the proposed TricorNet significantly outperforms the state of the art upon all three metrics. We found the best number of layers is two and the best convolution length is 30. We use $H = 64$ hidden states for each direction of the Bi-LSTM layers. We also observed that using the same number of hidden states for all the Bi-LSTM layers usually achieves better results, however, the number of hidden states has less influence on the results. To test the stability of performance, we use the same parameter setting to conduct multiple experiments. The results usually vary approximately within a 1% range. The parameters in each layer are randomly initialized; the same below.

**GTEA.** Table 2 shows that our proposed TricorNet achieves superior overlap F1 score, with a competitive frame-wise accuracy on GTEA dataset. The best accuracy is achieved by the ensemble approach from Singh et al. (2016b), which combines EgoNet features with TDD (Wang et al., 2015).

Table 2: Experimental results on Georgia Tech Egocentric Activities dataset. We use frame-wise accuracy (Acc.) and overlap F1 score with thresholds (F1@10,25,50) for evaluation. (See citations in Table 1 and text.)

| GTEA | Acc. | F1@{10, 25, 50} |
|---|---|---|
| Spatial CNN | 54.1 | 41.8, 36.0, 25.1 |
| ST-CNN | 60.6 | 58.7, 54.4, 41.9 |
| Bi-LSTM | 55.5 | 66.5, 59.0, 43.6 |
| EgoNet+TDD | **68.5** | - |
| Dilated TCN | 58.3 | 58.8, 52.2, 42.2 |
| ED-TCN | 64.0 | 72.2, 69.3, 56.0 |
| TricorNet (high) | 62.4 | 75.2, **71.3**, 58.0 |
| TricorNet (low) | 64.7 | **77.3, 73.4, 62.9** |
| TricorNet | **64.8** | **76.0**, 71.1, **59.2** |

Table 3: Experimental results on JHU-ISI Gesture and Skill Assessment Working Set dataset. We use frame-wise accuracy (Acc.) and segmental edit score (Edit) for evaluation. (See citations in Table 1 and text.)

| JIGSAWS | Acc. | Edit |
|---|---|---|
| MSM-CRF | 71.7 | - |
| Spatial CNN | 74.0 | 37.7 |
| ST-CNN | 77.7 | 68.0 |
| TCN | 81.4 | 83.1 |
| TricorNet (high) | 79.4 | 83.3 |
| TricorNet (low) | **82.2** | **84.9** |
| TricorNet | **82.9** | **86.8** |

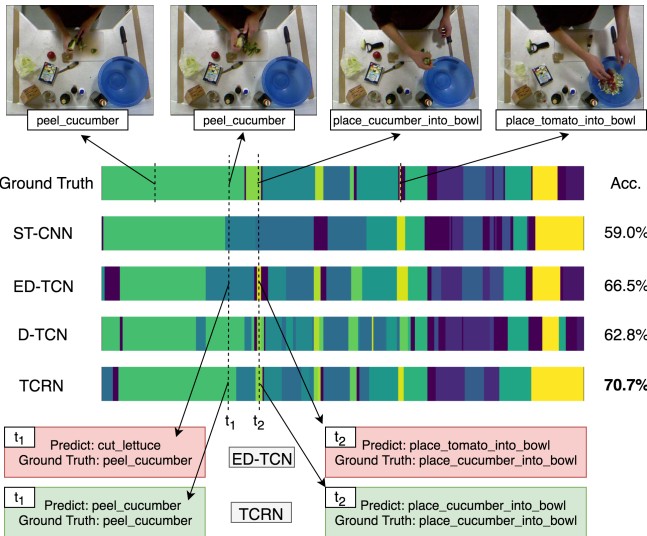

Figure 4: Top: Example images in a sample testing video from 50 Salads dataset. Middle: Ground truth and predictions from different models. Bottom: Two typical mistakes caused by visual similarity (left - peeled cucumber is similar to lettuce in color; right - hands will cover object when placing); TricorNet avoids the mistakes by learning long-range dependencies of different actions.

Since our approach uses simpler spatial features as Lea et al. (2017), it is very likely to improve our result by using the state-of-the-art features. Both TricorNet and TricorNet (low) have a good performance on all metrics and outperform ED-TCN. We use convolution length equals to 20 and $H = 64$ hidden states for LSTMs.

**JIGSAWS.** In Table 3, we show our state-of-the-art result on JIGSAWS dataset, where MSM-CRF is from Tao et al. (2013). Similar to the results on other datasets, TricorNet tends to achieve better segmental scores as well as keeping a superior or competitive frame-wise accuracy. Besides, TricorNet (low) also has a good performance, which may due to the relatively small size of the dataset. We set 15 for convolution length and 64 hidden states.

## 4.4 ACTION DEPENDENCIES

TricorNet is proposed to learn long-range dependencies of actions taking the advantage of hierarchical LSTM units, which is not a function of ED-TCN using only temporal convolution. Figure 4 shows the prediction results of a sample testing video from 50 Salads dataset. We find two typical mistakes made by other methods which are very likely due to visual similarity between different actions.

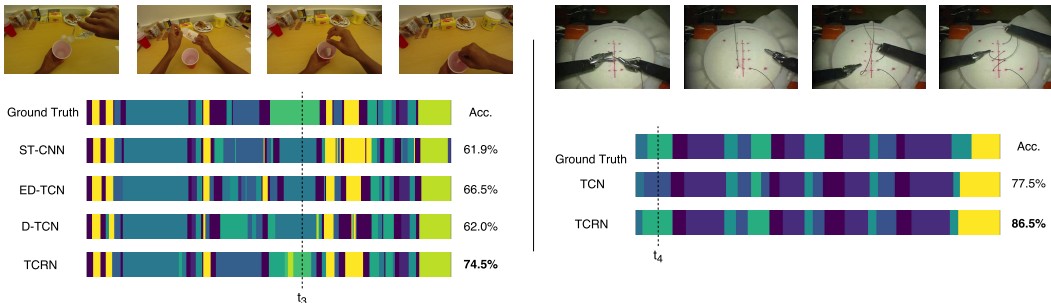

Figure 5: Left: Example images in a sample testing video from GTEA dataset, with ground truth and predictions from different models. Right: Example images in a sample testing video from JIGSAWS dataset, with ground truth and predictions from different models. Our proposed TricorNet avoids some classification mistakes when it can still perform a precise segmentation between actions.

ED-TCN predicts action at $t_1$ to be *cut lettuce*, of which the ground truth is *peel cucumber*. A possible reason is that the almost-peeled cucumber looks like lettuce in color. Therefore, when reaching some decision boundary before $t_1$, ED-TCN changes its prediction from *peel cucumber* to *cut lettuce*. However, TricorNet manages to preserve the prediction *peel cucumber* after $t_1$, possibly because of the overall rarity that *cut lettuce* comes directly after *peel cucumber*. In this case, TricorNet uses some long-range action dependencies help improve the performance of action segmentation.

At $t_2$, ED-TCN predicts action to be *place tomato into bowl*, which is very unreasonable to come after *peel cucumber*. A possible explanation is that when people placing objects, hands usually cover and make the objects hard to recognize. Thus, it is hard to visually distinguish between actions such as *place cucumber into bowl* and *place tomato into bowl*. However, TricorNet succeeds in predicting $t_2$ as *place cucumber into bowl* even it is hard to recognize from the video frames. It is possible that the dependency between *peel cucumber* and *place cucumber into bowl* is learned by TricorNet. This suggests that action dependency helps improve the accuracy of labeling a segmented part. We also find similar examples in GTEA and JIGSAWS datasets. Figure 5 shows that in some cases (at $t_3$ in GTEA and $t_4$ in JIGSAWS), TricorNet can achieve a better accuracy labeling segmented parts.

Thus, TricorNet can better learn the action ordering and dependencies, which will improve the performance of action segmentation when sometimes it is visually unrecognizable through video frames. This is very likely to happen in real world, especially for video understanding problems that the video data may be noisy or simply due to occlusions.

## 5 CONCLUSION

In this paper, we propose TricorNet, a novel hybrid temporal convolutional and recurrent network for video action segmentation problems. Taking frame-level features as the input to an encoder-decoder architecture, TricorNet uses temporal convolutional kernels to model local motion changes and uses bi-directional LSTM units to learn long-term action dependencies. We provide three model variants to comprehensively evaluate our model design. Despite the simplicity in methods, experimental results on three public action segmentation datasets with different metrics show that our proposed model achieves superior performance over the state of the art. A further qualitative exploration on action dependencies shows that our model is good at capturing long-term action dependencies, which help to produce segmentation in a smoother and preciser manner.

**Limitations.** In experiments we find that all the best results of TricorNet are achieved with number of layers $K = 2$. It will either over-fit or stuck in local optimum when adding more layers. Considering all three datasets are relatively small with limited training data (despite they are standard in evaluating action segmentation), using more data is likely going to further improve the performance.

**Future Work.** We consider two directions for the future work. Firstly, the proposed TricorNet is good to be evaluated on other action segmentation datasets to further explore its strengths and limitations. Secondly, TricorNet can be extended to solve other video understanding problems, taking advantage of its flexible structural design and superior capability of capturing video information.

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
