# OpenReview forum: "Video Action Segmentation with Hybrid Temporal Networks"
_ICLR.cc/2018/Conference — Reject_

### Official Review · AnonReviewer3 · 2017-11-26
**lacking in terms of novelty**

**Rating:** 4
**Confidence:** 4

**Review:**

The paper proposed a combination of temporal convolutional and recurrent network for video action segmentation. Overall this paper is written and easy to follow.

The novelty of this paper is very limited. It just replaces the decoder of ED-TCN (Lea et al. 2017) with a bi-directional LSTM. The idea of applying bi-directional LSTM is also not new for video action segmentation. In fact, ED-TCN used it as one of the baselines. The results also do not show much improvement over ED-TCN, which is much easier and faster to train (as it is fully convolutional model) than the proposed model. Another concern is that the number of layers parameter 'K'. The authors should show an analysis on how the performance varies for different values of 'K' which I believe is necessary to judge the generalization of the proposed model. I also suggest to have an analysis on entire convolutional model (where the decoder has 1D-deconvolution) to be included in order to get a clear picture of the improvement in performance due to bi-directional LSTM . Overall, I believe the novelty, contribution and impact of this work is sub-par to what is expected for publication in ICLR.

---

### Official Review · AnonReviewer1 · 2017-11-27
**Video Action Segmentation with Hybrid Temporal Networks**

**Rating:** 3
**Confidence:** 5

**Review:**

I will be upfront: I have already reviewed this paper when it was submitted to NIPS 2017, so this review is based heavily on the NIPS submission.

I am quite concerned that this paper has been resubmitted as it is, word by word, character by character. The authors could have benefited from the feedback they obtained from the reviewers of their last submissions to improved their paper, but nothing has been done. Even very easy remarks, like bolding errors (see below) have been kept in the paper.

The proposed paper describes a method for video action segmentation, a task where the video must be temporally densely labeled by assigned an action (sub) class to each frame. The method proceeds by extracting frame level features using convolutional networks and then passing a temporal encoder-decoder in 1D over the video, using fully supervised training.

On the positive side, the method has been tested on 3 different datasets, outperforming the baselines (recent methods from 2016) on 2 of them.

My biggest concern with the paper is novelty. A significant part of the paper is based on reference [Lea et al. 2017], the differences being quite incremental. The frame-level features are the same as in [Lea et al. 2017], and the basic encoder-decoder strategy is also taken from [Lea et al. 2017]. The encoder is also the same. Even details are reproduced, as the choice of normalized Relu activations.

The main difference seems to me that the decoder is not convolutional, but a recurrent network.

The encoder-decoder architecture seems to be surprisingly shallow, with only K=2 layers at each side.

The paper is well written and can be easily understood. However, a quite large amount of space is wasted on obvious and known content, as for example the basic equation for a convolutional layer (equation (1)) and the following half page of text and equations of LSTM and Bi-directional LSTM networks. This is very well known and the space can be used for more details on the paper's contributions.

While the paper is generally well written, there are a couple of exceptions in the form of ambiguous sentences, for example the lines before section 3.

There is a bolding error in table 2, where the proposed method is not state of the art (as indicated) w.r.t. to the accuracy metric.

To sum it up, the positive aspect of nicely executed experiments is contrasted by low novelty of the method.  To be honest, I am not totally sure whether the contribution of the paper should be considered as a new method or as architectural optimizations of an existing one. This is corroborated by the experimental results on the first two datasets (tables 2 and 3): on 50 salads, where ref. [Lea et al. 2017]. seems currently to obtain state of the art performance, the improvement obtained by the proposed method allows it to get state of the art performance. On GTEA, where [Lea et al. 2017] does not currently deliver state of the art performance, the proposed method performs (slightly) better than [Lea et al. 2017] but does not obtain state of the art performance.

On the third dataset, JIGSAWS, reference [Lea et al. 2017]. has not been tested, which is peculiar given the closeness.

---

### Official Review · AnonReviewer2 · 2017-11-27

**Rating:** 3
**Confidence:** 5

**Review:**

This paper discusses the problem of action segmentation in long videos, up to 10 minutes long. The basic idea is to use a temporal convolutional encoder-decoder architecture, where in the enconder 1-D temporal convolutions are used. In the decoder three variants are studied:

(1) One that uses only several bidirectional LSTMs, one after the other.
(2) One that first applies successive layers of deconvolutions to produce per frame feature maps. Then, in the end a bidirectional LSTM in the last layer.
(3) One that first applies a bidirectional LSTM, then applies successively 1-D deconvolution layer.

All variants end with a "temporal softmax"  layer, which outputs a class prediction per frame.

Overall, the paper is of rather limited novelty, as it is very similar to the work of Lea et al., 2017, where now the decoder part also has the deconvolutions smoothened by (bidirectional) LSTMs. It is not clear what is the main novelty compared to the aforementioned paper, other than temporal smoothing of features at the decoder stage.

Although one of the proposed architectures (TricorNet) produces some modest improvements, it is not clear why the particular architectures are a good fit. Surely, deconvolutions and LSTMs can help incorporate some longer-term temporal elements into the final representations. However, to begin with, aren't the 1-D deconvolutions and the LSTMs (assuming they are computed dimension-wise) serving the same purpose and therefore overlapping? Why are both needed?

Second, what makes the particular architectures in Figure 3 the most reasonable choice for encoding long-term dependencies, is there a fundamental reason? What is the difference of the L_mid from the 1-D deconv layers afterward? Currently, the three variants are motivated in terms of what the Bi-LSTM can encode (high or low level details).

Third, the qualitative analysis can be improved. For instance, the experiment with the "cut lettuce" vs "peel cucumber" is not persuasive enough. Indeed, longer temporal relationships can save incorrect future predictions. However, this works both ways, meaning that wrong past predictions can persist because of the long-term modelling. Is there a mechanism in the proposed approach to account for that fact?

All in all, I believe the paper indeed improves over existing baselines. However, the novelty is insufficient for a publication at this stage.

---

### Decision · Program_Chairs · 2018-01-29
**ICLR 2018 Conference Acceptance Decision**

**Decision:**

Reject

**Comment:**

All reviewers believed that the novelty of the contribution was limited.